# Removal of Fish Odors Form Styrofoam Packaging to Improve Recycling Potential Using Hansen Solubility Parameters

**Takeshi Ishida**

Department of Ocean Mechanical Engineering, National Fisheries University, Shimonoseki 759-6595, Japan; ishida07@ecoinfo.jp

**Abstract:** Styrofoam fish containers (fish boxes) are ideal for transporting fresh fish because of their light weight and insulation properties. However, due to fish-like odors, fish boxes are simply thrown out after use and are limited to low-grade recycling in Japan. To improve their recyclability, we investigated trimethylamine, which causes fish-like odor, to ascertain whether it is soluble in vegetable oil using the Hansen solubility parameter (HSP). At present, the Oshima College method (OCMT), which is used to reduce the volume of styrofoam, uses heated vegetable oil and can potentially remove the fish-like odor. In addition, the solubility of dimethyl sulfide, which causes the sea-like smell in styrofoam found drifting on shores, in vegetable oil was investigated. Our results conclude that OCMT can remove the fish- and sea-like odors found in waste styrofoam and thus improve its recycling potential.

**Keywords:** styrofoam; trimethylamine; dimethyl sulfide; vegetable oil; Hansen solubility parameter

## 1. Introduction

Styrofoam, which is expanded polystyrene foam, is used in containers such as fish boxes (Figure 1). Its properties, such as its light weight and excellent insulation properties, make it ideal for transporting fresh fish. Although there is limited information on the production volume of fish boxes, according to the Japan Expanded Polystyrene Association (JEPSA), in 2009, the volume of containers for agricultural and marine use was 91,300 tons [1] and the shipment volume of polystyrene was 141,000 tons [2] in Japan. Styrofoam for fish and marine products accounts for 55.8% of styrofoam shipments, of which 70% of the containers are used to transport fish. At present, for hygienic purposes, fish boxes are used only once, thus generating a large volume of discarded fish boxes in Japan. For this reason, in large-scale markets, such as the Tokyo Metropolitan Central Wholesale Market, devices are installed to compress the fish boxes into plastic ingots. As fish boxes are used only for fresh food transportation, they are free of harmful substances and are suitable for recycling. However, fish-like odors limit the ingots to low-grade recycling. The ingots are only used as fuel in thermal recycling or as admixture in the production of styrene products for building materials [3,4]. In recent years in Japan, the remaining ingots have been exported to China and other Asian countries. However, China banned the import of plastic waste in December 2017 [5]. Some waste plastics have been exported to other Asian countries, but they are also banning the import of plastic waste. Hence, a recycling process for waste fish boxes in Japan is urgently required.

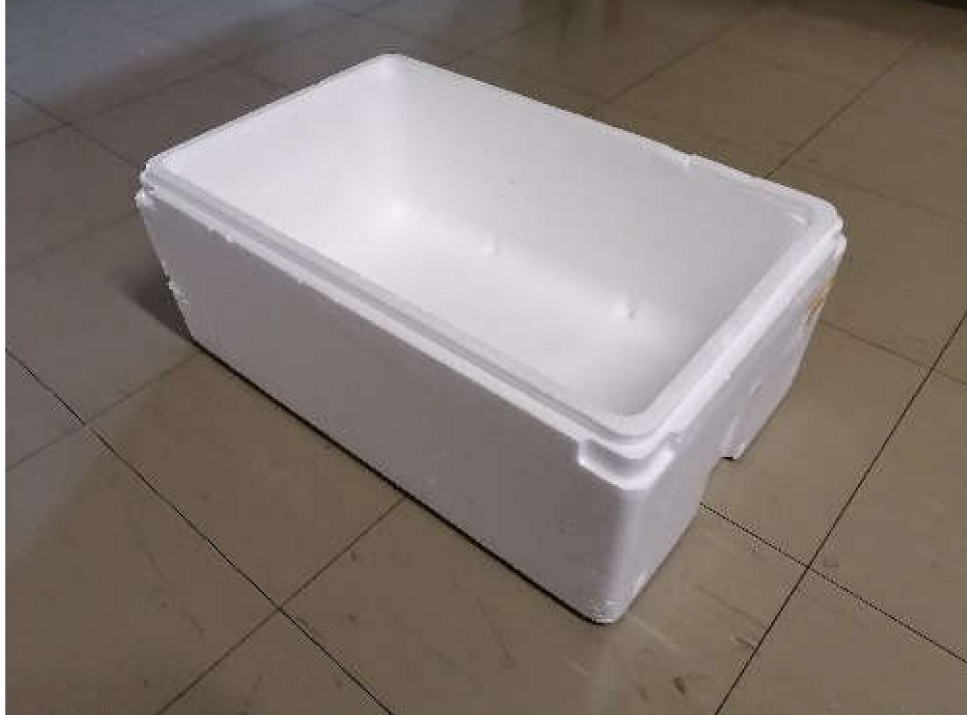

**Figure 1.** Styrofoam fish container, commonly known as a fish box.

One of the factors hindering the material recycling of discarded fish boxes is the fish-like odor. Fish boxes are manufactured by filling polystyrene beads treated with a foaming agent into a mold and expanding the foaming agent under high-pressure and high-temperature conditions, creating aggregate cells of expanded beads welded together. The fish-like odor permeates and adheres to the gap between the cells, and it is difficult to remove it with water. Methods for deodorization of fish boxes before recycling include a neutralization method using a dilute hydrochloric acid solution and a masking method using limonene extracted from citrus peel. However, these methods have drawbacks, such as the necessity of treating the used solution with waste liquid, and no practical method has been established.

The smell is caused by the reduction of trimethylamine oxide (($CH_3$)$_3$NO) by bacteria, producing trimethylamine (($CH_3$)$_3$N) [6]. Trimethylamine is an alkaline volatile substance and can be removed by baking or neutralization with an acid. Notably, it is soluble in organic solvents. In addition, various other fats and oils in fish also cause an odor when oxidized.

At present, the following techniques are used for treating waste styrofoam: (1) styrofoam oiling equipment (equipment for producing styrene oil from styrofoam), (2) desalting and volume reduction technology using heated waste vegetable oil, (3) a melting treatment with an organic solvent, and (4) processing using a shear compression crusher.

The first method uses direct heat to convert styrofoam to styrene oil [7] and is used in Japan to treat styrofoam waste found drifting along the coast. There are pilot projects involving this method in Japan such as Tsushima Island in Nagasaki Prefecture and Hatoma Island in Okinawa Prefecture.

The second method, also used for treating styrofoam waste drifting on the coast, is a technique for desalting and reducing the volume using heated vegetable oil. Kawahara et al. [8] of the National Institute of Technology, Oshima College, reported that the volume could be reduced to approximately 1/50th and can be solidified to produce plastic ingots. This volume reduction method, called the Oshima College method (OCMT), uses waste vegetable oil as a solvent [9]. Styrofoam is heated to 160–200 °C with oil as a solvent to expel the air in styrofoam. By heating, the trapped air expands, and the structure of the styrene foam also becomes soft simultaneously, which causes the air to be discharged to the outside. The cells of styrofoam are destroyed and the volume is reduced (Figures 2 and 3). In addition, it was reported

that the salt attached to the surface of the polystyrene foam was desorbed, and a desalting effect was observed [8].

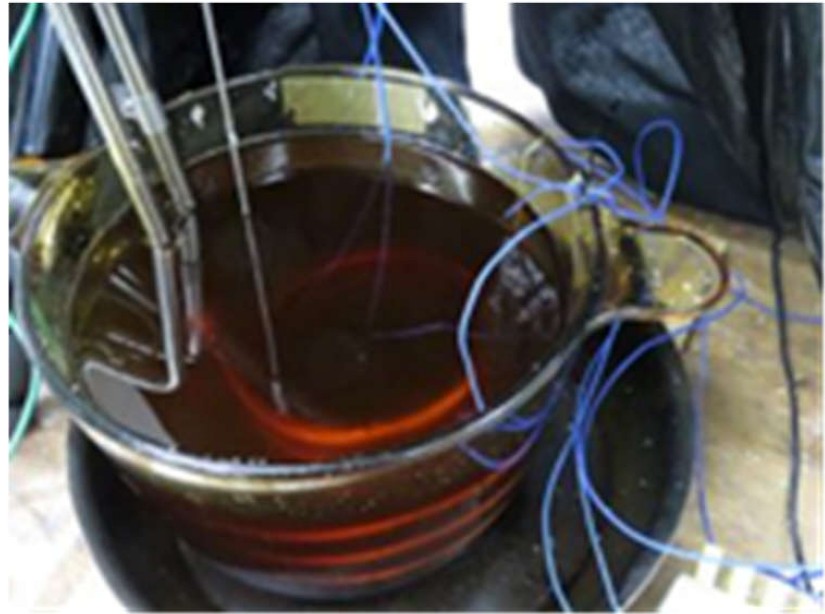

**Figure 2.** Experimental equipment for OCMT (Oshima College method), which is a technique for desalting and reducing the volume using heated vegetable oil (image provided by Kawahara).

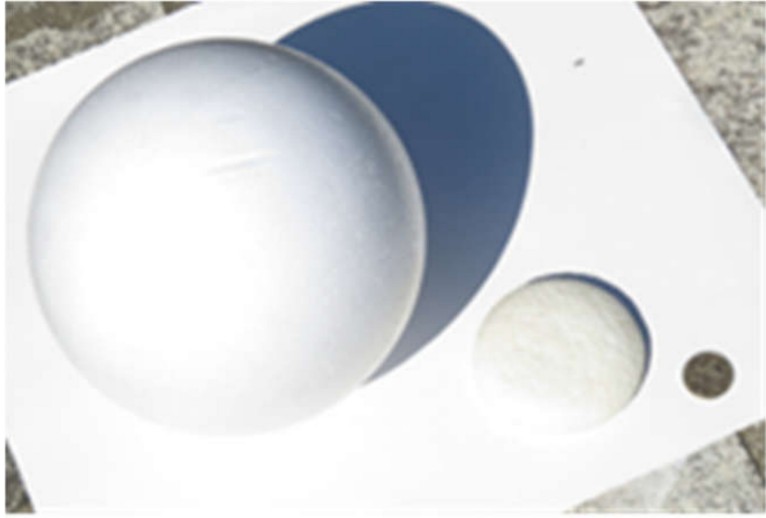

**Figure 3.** Reduction in the volume of styrofoam (original diameter, 200 mm) and a Japanese 500 yen coin (image provided by Kawahara).

The third method is to dissolve styrofoam into an organic solvent at room temperature [10] and is used for processing food containers in retail stores. However, the organic solvent used requires additional waste treatment.

The fourth method is a compression crusher and is used in wholesale markets to reduce the volume of containers transporting farm and marine products. One example of a compressor is STYROS, from Japanese company (ELCOM Co., Ltd., Sapporo City, JAPAN)[11]. The cells in the polystyrene foam are broken with heat from compression, and polystyrene foam melts and reduces volume; however, the attached fish-like odor does not evaporate and stays in the polystyrene ingot. This makes it difficult to remove the components with a fish-like odor further downstream in the waste treatment process.

In this study, we investigated the solubility of trimethylamine in vegetable oil using the Hansen solubility parameter (HSP) to confirm whether OCMT can remove the fish-like odor. It is known that trimethylamine is soluble in various organic solvents, such as toluene, but its solubility in vegetable oil is not known. If trimethylamine can be easily desorbed from the styrofoam surface using vegetable oil, OCMT performs a deodorizing function in addition to volume reduction and desalination. To our knowledge, there have been no reports on methods for deodorizing trimethylamine or dimethyl sulfide from polystyrene. Although there is a report on the deodorizing effect of Eriobotrya Japonica seed oil [12], there is no literature on the solubility of trimethylamine and dimethyl sulfide in vegetable oils.

## 2. Method

### 2.1. Overview of theHansen Solubility Parameter

In this study, the solubility of trimethylamine in vegetable oils was evaluated using HSP. HSP is a measure of the solubility of a substance in a solvent and was proposed by Charles M. Hansen in 1967 [13]. HSP is based on the idea that two substances with similar interactions between molecules will be easily miscible with one another. In order to evaluate these interactions, three parameters are used: $\delta_d$, which is the energy owing to intermolecular dispersion forces (van der Waals forces); $\delta_p$, which is the energy owing to dipole interactions between molecules; and $\delta_h$, which is the energy owing to hydrogen bonding between molecules. The units for the parameters are $MPa^{0.5}$. These parameters can be represented by a specific vector, unique to each substance, which can be represented in a three-dimensional space, the so-called Hansen space. Two substances with two vectors having similar direction and length are considered to have high solubility. First, the distance between the solute HSP and the solvent HSP is calculated using Equation (1):

$$(Ra)^2 = 4(\delta_{d2} - \delta_{d1})^2 + (\delta_{p2} - \delta_{p1})^2 + (\delta_{h2} - \delta_{h1})^2 \tag{1}$$

where Ra is the vector for the solubility parameters of the solvent and solute. Furthermore, the solute has a specific interaction radius, $R_0$. If the tip of the vector for the solvent HSP exists inside the sphere with a radius $R_0$, wherein the center is the tip of the solute HSP vector, the solute can be determined to be soluble in the solvent (Figure 4). $R_0$ can be estimated by conducting experiments to determine whether a number of solvents are soluble or insoluble. The relative energy difference (RED) is the ratio of Ra and $R_0$. If RED < 1 (HSP of the solvent is inside the sphere of radius $R_0$), the solute is expected to be soluble, partially soluble if RED = 1 (HSP of the solvent is on the surface of the sphere), and insoluble if RED > 1 (HSP of the solvent is outside the sphere).

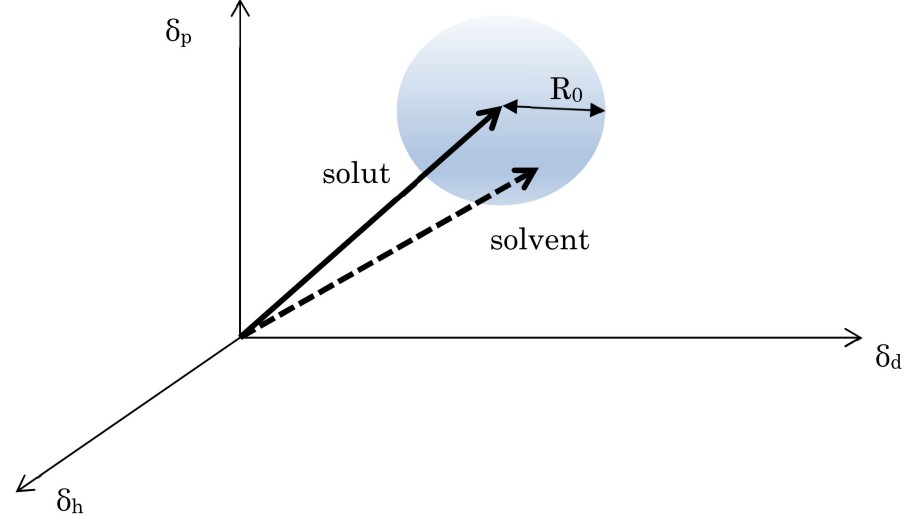

**Figure 4.** Outline of Hansen solubility parameter and interaction radius $R_0$.

## 2.2. Evaluation of Substances

In this study, the following substances were evaluated using HSP. Three substances were evaluated as solutes: trimethylamine (fish-like odor), dimethyl sulfide (sea-like odor generated by the decomposition of plankton and seaweed) [14], and polystyrene. Styrofoam used the parameters of polystyrene in this study. The CAS (Chemical Abstracts Service) registration numbers were 9003-53-6, 75-50-3, and 75-18-3, respectively. Vegetable oil was chosen as the solvent due to its use in OCMT. The following vegetable oils were investigated: oleic acid (CAS registration number 112-80-1), linoleic acid (CAS registration number 60-33-3), stearic acid (CAS registration number 57-11-4), palmitic acid (CAS registration number 57-10-3), and limonene (CAS registration number 138-86-3). Although limonene is not used in OCMT, it dissolves ttyrofoam [15], and therefore, it was included in the evaluation for comparison.

OCMT uses used vegetable oil, and the actual vegetable oil is a triacylglycerol. A triacylglycerol (triglyceride) has three different fatty acids that are bonded through an ester to glycerol. Various triacylglycerols exist depending on the types and combinations of the fatty acids to be bound. In this study, the following triacylglycerols were evaluated: Tripalmitin (CAS registration number 555-44-2), 1-*O*-Palmitoyl-2-*O*-linoleoyl-3-*O*-stearoyl-glycerol, and 1-*O*-Linoleoyl-2-*O*-palmitoyl-3-*O*-stearoyl-glycerol.

## 2.3. Evaluation Method

HSPs in Practice (HSPiP) is an integrated packaged software for Windows developed to evaluate HSPs and includes a database and an e-book. The value for $R_0$ of polystyrene is listed in the HSPiP database and was used in the evaluation. However, $R_0$ values for trimethylamine and dimethyl sulfide are not listed in the database. Normally, determining $R_0$ by examining the solubility using various solvents is necessary. However, in this study, a provisional value of $R_0$ was determined from the fact that RED < 1 if the organic solvent is known to reliably dissolve a solute. Although the actual $R_0$ may be larger than the estimated $R_0$, if the RED value of each solvent is ≤1, it can be determined that the solvent is reliably dissolved. HSP parameters of triacylglycerol are also not listed in the HSPiP database. Therefore, estimated values using the chemical structural formula were obtained using the estimation function of HSPiP [13].

## 3. Results

### 3.1. Identifying Hansen Solubility Parameters

Tables 1 and 2 show HSPs of each solute and solvent obtained from the database of HSPiP. Furthermore, the $R_0$ value for polystyrene was obtained from the database. Table 3 shows the parameters of the organic solvent used for estimating the value of $R_0$ for trimethylamine and dimethyl sulfide. HSP parameters of triacylglycerol are not listed in the HSPiP database; therefore, estimated values were obtained using the estimation function of HSPiP (Table 4). The detailed estimation method is shown in Chapter 23 of the Hansen Handbook [6].

**Table 1.** Hansen solubility parameters of each solute.

| Solute | CAS | $\delta_d$ | $\delta_p$ | $\delta_h$ | $R_0$ |
|---|---|---|---|---|---|
| Polystyrene | 9003-53-6 | 22.3 | 5.75 | 4.3 | 12.7 |
| Teimethyl amine | 75-50-3 | 14.6 | 3.4 | 1.8 | - |
| Dimethyl sulfide | 75-18-3 | 16.1 | 6.4 | 7.4 | - |

**Table 2.** Hansen solubility parameters of each solvent.

| Solvent | CAS | $\delta_d$ | $\delta_p$ | $\delta_h$ |
|---------|-----|------------|------------|------------|
| Oleic acid | 112-80-1 | 16.0 | 2.8 | 6.2 |
| Linoleic acid | 60-33-3 | 16.7 | 3.1 | 6.1 |
| Stearic acid | 57-11-4 | 16.3 | 3.3 | 5.5 |
| Palmitic acid | 57-10-3 | 16.2 | 3.3 | 5.8 |
| Limonene | 138-86-3 | 17.2 | 1.8 | 4.3 |

**Table 3.** Hansen solubility parameters of each organic solvent.

| Solvent | CAS | $\delta_d$ | $\delta_p$ | $\delta_h$ |
|---------|-----|------------|------------|------------|
| Benzene | 71-43-2 | 18.4 | 0.0 | 2.0 |
| Acetone | 67-64-1 | 15.5 | 10.4 | 7.0 |
| Toluene | 108-88-3 | 18.0 | 1.4 | 2.0 |

**Table 4.** Estimated value of Hansen solubility parameter of each triacylglycerol.

| Solvent | CAS | $\delta_d$ | $\delta_p$ | $\delta_h$ |
|---------|-----|------------|------------|------------|
| Tripalmitin | 555-44-2 | 16.4 | 1.8 | 1.9 |
| 1-*O*-Palmitoyl-2-*O*-linoleoyl-3-*O*-stearoylglycerol | - | 16.53 | 1.66 | 2.96 |
| 1-*O*-Linoleoyl-2-*O*-palmitoyl-3-*O*-stearoylglycerol | - | 16.54 | 1.54 | 2.81 |

### 3.2. Solubility of Polystyrene in Vegetable Oil

Table 5 shows the RED values of polystyrene for the various vegetable oils investigated. For each of the vegetable oils, the RED value is around 1; thus, it can be predicted that polystyrene is not soluble in the chosen vegetable oils. However, since the values are so close to 1, there may be some solubility. The RED value of limonene was 0.86, indicating that polystyrene was soluble in limonene. This is consistent with experimental data that show that styrofoam is generally soluble in limonene.

**Table 5.** Relative energy difference (RED) of polystyrene for vegetable oils.

| Solvent | $R_0$ of Polystyrene | Ra | Red Number = Ra/$R_0$ |
|---------|----------------------|-----|-----------------------|
| Oleic acid | 12.7 | 13.1 | 1.03 |
| Linoleic acid | 12.7 | 11.6 | 0.92 |
| Stearic acid | 12.7 | 12.3 | 0.97 |
| Palmitic acid | 12.7 | 12.5 | 0.99 |
| Limonene | 12.7 | 10.9 | 0.86 |
| Tripalmitin | 12.7 | 12.7 | 1.00 |
| 1-*O*-Palmitoyl-2-*O*-linoleoyl-3-*O*-stearoylglycerol | 12.7 | 12.3 | 0.97 |
| 1-*O*-Linoleoyl-2-*O*- palmitoyl-3-*O*-stearoylglycerol | 12.7 | 12.4 | 0.97 |

### 3.3. Solubility of Trimethylamine in Vegetable Oil

To evaluate the solubility of trimethylamine in vegetable oil, estimating $R_0$ for trimethylamine is necessary. Normally, it is necessary to determine $R_0$ by examining the solubility using various solvents. However, herein, a provisional value for $R_0$ was determined based on the fact that RED < 1 when trimethylamine is soluble in an organic solvent.

Organic solvents were selected in which trimethylamine is soluble. Table 6 shows the RED values for trimethylamine and the organic solvents. $R_0$ is a given value at which the RED value becomes

almost 1. When the $R_0$ value is 9, it can be seen that the RED value of each organic solvent is <1. The actual $R_0$ value of trimethylamine may be higher than 9; if the RED value calculated for a certain solvent with $R_0 = 9$ has a value of 1 or less, it is certain that trimethylamine will have a similar Hansen parameter than the organic solvents, and it can be determined that they are soluble in the solvent.

**Table 6.** Relative energy difference (RED) of trimethylamine for organic solvent.

| Solvent | $R_0$ | Ra | Red Number = Ra/$R_0$ |
|---------|-------|-----|------------------------|
| Benzene | 9 | 8.3 | 0.93 |
| Acetone | 9 | 8.9 | 0.99 |
| Toluene | 9 | 7.1 | 0.79 |

Table 7 shows the RED value of trimethylamine in each vegetable oil when calculated with $R_0 = 9$. Since all RED values are ≤1, it can be determined that trimethylamine is soluble in each vegetable oil.

**Table 7.** Relative energy difference (RED) of trimethylamine for each vegetable oil.

| Solvent | $R_0$ of Trimethyl Amine | Ra | Red Number = Ra/$R_0$ |
|---------|--------------------------|-----|------------------------|
| Oleic acid | 9 | 5.2 | 0.58 |
| Linoleic acid | 9 | 6.0 | 0.67 |
| Stearic acid | 9 | 5.0 | 0.56 |
| Palmitic acid | 9 | 5.1 | 0.57 |
| Limonene | 9 | 6.0 | 0.67 |
| Tripalmitin | 9 | 3.9 | 0.44 |
| 1-*O*-Palmitoyl-2-*O*-linoleoyl-3-*O*-stearoylglycerol | 9 | 4.4 | 0.49 |
| 1-*O*-Linoleoyl-2-*O*-palmitoyl-3-*O*-stearoylglycerol | 9 | 4.4 | 0.49 |

### 3.4. Solubility of Dimethyl Sulfide in Vegetable Oil

Similarly, using the knowledge that dimethyl sulfide is soluble in certain organic solvents, the result estimating the lower limit of the $R_0$ value was 8.4, as shown in Table 8. Using this value, the RED values for dimethyl sulfide in the chosen vegetable oils were estimated, as shown in Table 9. The RED value for each vegetable oil was ≤1, indicating that dimethyl sulfide is soluble in the vegetable oils. Moreover, it was found that dimethyl sulfide may not be soluble in limonene because the RED value is greater than 1.

**Table 8.** Relative energy difference (RED) of dimethyl sulfide for each organic solvent.

| Solvent | $R_0$ | Ra | Red Number = Ra/$R_0$ |
|---------|-------|-----|------------------------|
| Diethyl ether | 8.4 | 5.5 | 0.66 |
| acetone | 8.4 | 4.2 | 0.50 |
| toluene | 8.4 | 8.3 | 0.99 |

**Table 9.** Relative energy difference (RED) of dimethyl sulfide for each vegetable oil.

| Solvent | $R_0$ of Dimethyl Sulfide | Ra | Red Number = Ra/$R_0$ |
|---------|---------------------------|-----|------------------------|
| Oleic acid | 8.4 | 3.8 | 0.45 |
| Linoleic acid | 8.4 | 3.7 | 0.45 |
| Stearic acid | 8.4 | 3.7 | 0.44 |
| Palmitic acid | 8.4 | 3.5 | 0.42 |
| Limonene | 8.4 | 6.0 | 0.71 |
| Tripalmitin | 8.4 | 7.2 | 0.86 |
| 1-*O*-Palmitoyl-2-*O*-linoleoyl-3-*O*-stearoylglycerol | 8.4 | 6.6 | 0.78 |
| 1-*O*-Linoleoyl-2-*O*-palmitoyl-3-*O*-stearoylglycerol | 8.4 | 6.7 | 0.80 |

## 4. Discussion

From the evaluation by HSP, it was found that trimethylamine and dimethyl sulfide are sufficiently soluble in various types of vegetable oil. However, OCMT mainly uses used frying oil, for which it is difficult to estimate HSPs because used frying oil is a mixture of various oils and fats. However, HSP estimates of used frying oil, based on experiments, were obtained by Batista et al. [16]. The values were found to be $\delta_d$ = 15.35, $\delta_p$ = 3.77, and $\delta_h$ = 6.87, which are not significantly different from the fatty acids used in the evaluation of this study (Table 4). When the RED value was calculated from these values, it was 0.59 for trimethylamine and 0.63 for dimethyl sulfide, which are both less than 1. Therefore, it can be determined that they are soluble in used frying oil. However, the evaluation by HSPs is only an estimated value, and it is necessary to experimentally confirm whether it is soluble.

## 5. Conclusions

In this study, the results obtained using HSPs predict that trimethylamine and dimethyl sulfide can be dissolved in vegetable oil. These results show that OCMT is able to perform the function of deodorizing waste styrofoam found on the coast and used for fish boxes in wholesale markets. This is in addition to desorbing the salt attached to the surface of the polystyrene foam. To confirm the results of this paper, experimental data are needed to ensure the evaluation values by calculation. The planning and preparing for such an experiment shall be conducted in a further study.

Furthermore, the deodorized fish boxes are more likely to be recycled into high-grade raw materials, and the vegetable oil used as a solvent can be converted into biodiesel fuel. In the future, as shown in Figure 5, it will be possible to construct a closed system that does not generate waste liquid. Regarding concerns about the effects of amines from trimethylamine in biodiesel oil, there are cases where amines are used as antioxidants for biodiesel [17], and it is thought that mixing of amines is not a problem.

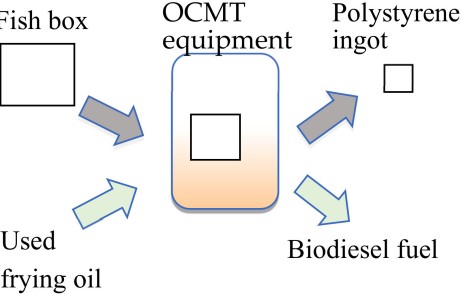

**Figure 5.** Volume reduction and recycling process of styrofoam using vegetable oil. The deodorized fish boxes are more likely to be recycled into high-grade raw materials, and the vegetable oil used as a solvent can be converted into biodiesel fuel. In the future, it will be possible to construct a closed system that does not generate waste liquid.

In addition, a large amount of waste can be found drifting on the shores of the islands facing the East China Sea, such as Tsushima in Nagasaki, Japan. Most of this waste is styrofoam-based, and if the proposed process is realized, it is thought that it can contribute to the treatment of this waste. The cause of the sea-like odor attached to the drifting polystyrene foam is dimethyl sulfide, and it is necessary to experimentally evaluate its deodorizing effect in the future. This will open the possibility of high-quality recycling for drifting styrofoam, which will contribute to the environmental protection of the oceans and coastlines.

**Funding:** This research received no external funding.

**Acknowledgments:** The author would like to thank Hideo Kawahara (National Institute of Technology, Oshima College, Japan) for providing images and data on the OCMT method. The author would also like to thank Enago (www.enago.jp) for the English language review.

**Conflicts of Interest:** The author declares no conflict of interest.

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
