# Peer review of "Removal of Fish Odors Form Styrofoam Packaging to Improve Recycling Potential Using Hansen Solubility Parameters"

_recycling, doi:10.3390/recycling5040030_

Round 1
Reviewer 1 Report
The manuscript is presenting investigation the solubility of trimethylamine in vegetable oil using the Hansen solubility parameter (HSP) to confirm whether OCMT can remove the fish-like odor.
The manuscript is of low scientific quality, methodology should be improved and discussion should be explained in details.
Author Response
Thank you for your peer review. I made some additions based on the suggestions of other reviewers. All the revised points are highlighted in the revised manuscript.
Reviewer 2 Report
The paper is clearly written and discusses a topic of relevance for the plastic recycling community.
As a general remark, some more references should be provided to support the introduction. The discussion of the different treatment techniques is not detailed or unclear in some points
To cite some examples:
line 34-35: The ingots are only used as fuel in thermal recycling... Reference?
line 36: China banned the import of plastic waste in December 2017... Reference?
lines 43-55: some reference should support the description of the causes of the fishy/sea odor. In particular, the role of trimethylamine and dimethyl sulfide should be highlighted
lines 60-79: only method 2 is described in some detail and with references.
line 68: "to expand the air in styrofoam" is not clear, the heated oil is collapsing the foam structure or the expansion of entrapped gases is responsible for it?
line 77: "to form a gel" without any solvent? A gel is typically a two-phase system
Some other points to be clarified:
line 106: Hansen parameters are not exactly energies (they are indeed axpressed in MPa^0.5)
line 120: "solvents are soluble or unsoluble" please revise
Table 8 ethanol is also included, showing RED > 1. Is Ethanol a solvent for dimethyl sulfide?
As a final point, it would be very useful to have an idea of the fate of waste oils used for styrofoam treatment in the OCMT method. In the conclusion section, a conversion into biodiesel is suggested: could the presence of dissolved amine (or other contaminants released by the waste styrofoam) be of concern for this conversion?
Reviewer 3 Report
The presented work is truly worthy of attention and publication since it is concerned with the most important problem of processing of styrofoam, in particular, containers for transporting of fresh fish and marine products. This problem is hot topic not only for Japan, but for all regions and countries with advanced fishing industry because such containers are disposable and generate a huge amount of difficult-to-recycle waste (due to presence of fish-like and sea-like odors). The article provides a good overview of styrofoam waste treatment methods which are currently in use. For the first time an author has carried out an evaluative study of the solubility of trimethylamine and dimethyl sulfide (odors-generating substances) in used vegetable oil using the Hansen solubility parameter. Additionally, the solubility of polystyrene in various vegetable oils was evaluated. The essence of the method is described in detail and clearly, a reference to the primary source is given. The data obtained are beyond doubt and are of undoubted practical value for implementation into processing of styrofoam with fish-like or sea-like odors using vegetable oil as odor-removing agent. In my opinion, the inclusion of experimental data in the article would significantly increase its importance for the industry, nevertheless, I definitely recommend publishing the article in its current form.
I have only one comment: in the sections Acknowledgments and Conflicts of Interest, the author writes about himself in a plural. The word "authors" should be replaced by "author".
Round 2
Reviewer 2 Report
The manuscript has been improved and the requested information has been added. It can be now accepted for publication.